# The Role of Telomerase in Breast Cancer’s Response to Therapy

**DOI:** 10.3390/ijms232112844

**Published:** 2022-10-25

**Authors:** Eliza Judasz, Natalia Lisiak, Przemysław Kopczyński, Magdalena Taube, Błażej Rubiś

**Affiliations:** 1Department of Clinical Chemistry and Molecular Diagnostics, Poznan University of Medical Sciences, 60-806 Poznan, Poland; 2Centre for Orthodontic Mini-Implants at the Department and Clinic of Maxillofacial Orthopedics and Orthodontics, Poznan University of Medical Sciences, 60-812 Poznan, Poland

**Keywords:** telomerase, breast cancer, therapy, resistance

## Abstract

Currently, breast cancer appears to be the most widespread cancer in the world and the most common cause of cancer deaths. This specific type of cancer affects women in both developed and developing countries. Prevention and early diagnosis are very important factors for good prognosis. A characteristic feature of cancer cells is the ability of unlimited cell division, which makes them immortal. Telomeres, which are shortened with each cell division in normal cells, are rebuilt in cancer cells by the enzyme telomerase, which is expressed in more than 85% of cancers (up to 100% of adenocarcinomas, including breast cancer). Telomerase may have different functions that are related to telomeres or unrelated. It has been shown that high activity of the enzyme in cancer cells is associated with poor cell sensitivity to therapies. Therefore, telomerase has become a potential target for cancer therapies. The low efficacy of therapies has resulted in the search for new combined and more effective therapeutic methods, including the involvement of telomerase inhibitors and telomerase-targeted immunotherapy.

## 1. Introduction

Cancer is still one of the most dangerous diseases of the 21st century, and the most common type in women worldwide is breast cancer [1,2]. The progress in the field of molecular biology in recent decades enabled understanding the molecular basis of cancer and provided significant progress in cancer diagnostics and therapy. However, due to increasing environmental risk factors and the aging of societies, the incidence numbers continue to rise. Over the past three decades, scientific interest in the field of cancer cell immortality related to telomerase and telomeres has increased significantly. In 2009, the Nobel Prize in Physiology and Medicine was awarded to three scientists, E. Blackburn, J. Szostak, and C. Greider, for “the discovery of how the ends of chromosomes are protected by structures called telomeres and the enzyme telomerase” [3]. This knowledge has allowed us to understand some of the mechanisms involved in cancer cells and to plan promising new treatments. Importantly, the idea of how telomerase contributes to cancer cell metabolism has significantly extended since that time, suggesting important roles of this enzyme in telomere-unrelated mechanisms. Progressive studies have shown that the role of telomerase extends beyond mechanisms associated with telomere restoration, e.g., resistance to therapy, adhesion, migration, etc. [4]. We hope that proceedings in this area will provide the development of novel and effective oncological therapy strategies. A telomerase-based therapeutic approach may be particularly efficient in breast cancer cells that, similar to other adenocarcinomas, abundantly express this enzyme [4].

## 2. Breast Cancer—A Challenge

Breast cancer originates from the epithelial tissue of the ducts or lobules of the mammary gland and primarily develops locally in the breast [5]. Cancer cells are capable of translocating due to a very primitive but effective amoeboid mechanism [6]. Breast cancer cells acquire a very dynamic characteristic that is associated with tumor cells’ ability to induce the epithelial–mesenchymal transition (EMT) and, conversely, the mesenchymal–epithelial transition (MET) [7]. Consequently, by changing their phenotype, cells are able to escape the bulk, enter the cardiovascular system, and metastasize to lymph nodes and distant organs such as bones, the liver, or the brain [7,8]. The ability of breast cancer cells to metastasize is the main cause for treatment failure and patient deaths. However, the mechanisms that contribute to this phenomenon are not yet fully understood. Different metabolic phenotypes of breast cancer cells are modulated by both intrinsic (such as *MYC* amplification, *PIK3CA*, and *TP53* mutations) and extrinsic factors (such as hypoxia, oxidative stress, and acidosis) [9]. The pathways involved in the modulation of breast cancer cell mobility are associated with the metalloproteinases MMP1 and ADAMTS1, which drive metastasis to bones. In turn, when metastasis to the lungs is reported, the combinatorial effects of *COX2*, *EREG* (*cyclooxygenase-2* and epiregulin, respectively), and *MMP1* and *2* are raised [10]. Importantly, the pharmacological inhibition of pathways mediated by these factors was shown to diminish the aggressiveness of the metastatic phenotypes [11]. Thus, as the EMT accounts for a broad spectrum of transitional stages between the epithelial and mesenchymal phenotypes, it is critical to find a common feature of all these stages. Apart from the modulation of the mentioned pathways, some specific genes and proteins are directly involved in adhesion and cell-to-cell interactions, including α1 and β1 integrins, cadherins, selectins, focal adhesion kinase (FAK), Akt, and other cell adhesion molecules as well as epigenetic factors, including miRNA [12]. Importantly, one of the most characteristic features of the vast majority of adenocarcinomas, including breast cancer, is the restoration of the telomerase expression (specifically the key catalytic telomerase subunit hTERT) and activity that provide tumor cell immortality. Telomerase is also postulated to modulate cancer cells’ responses to drugs as well as their metastatic potential via affecting adhesion pathways [13]. For these reasons, telomerase seems to be a good therapeutic target in breast cancer.

## 3. Obstacles in Therapeutic Strategies

Breast cancer can be treated locally or systemically. The choice of treatment method should be made by a multidisciplinary team of specialists in consultation with the patient and should be based on clinical and pathomorphological assessments, including the grade of malignancy, the histological type, ER/PgR, Ki67, and HER2 expression, the presence and location of metastases, etc. Treatment may include surgical treatment (sometimes including mastectomy), radiotherapy, chemotherapy with alkylating drugs (anthracyclines and taxoids), hormone therapy, and subsequent rehabilitation [14]. The increasing effectiveness of cancer treatment is due to better and earlier diagnostics supported by molecular markers and the screening of therapeutic targets. In cancer cells, many mechanisms are altered, including cell death inhibition (apoptosis suppression), changed drug metabolism, epigenetic modulation, changes in drug targets, enhanced DNA repair, and gene amplification that may all lead to resistance to therapy [15]. Some of the key players in drug resistance development are ATP-binding cassette (ABC) family proteins. However, as DNA stability is also critical for the process, telomeres and telomerase (specifically the catalytic telomerase subunit hTERT) are also suggested to contribute to the drug-resistant phenotype of cancer and the therapy response [16]. 

## 4. Immortality of Cancer Cells

Cancer cells share a common set of characteristics that distinguish them from normal cells. They allow tumor formation and provide the possibility of infiltration and metastasis to other organs. Although cancer cells retain the structural and functional features of the cells from which they were formed, during the process of tumor formation they acquire, among others, the following features: unlimited proliferation potential, independence from external growth signals, self-sufficiency in growth factors, resistance to proapoptotic signaling, the ability to generate angiogenesis, infiltration, and metastasis [17,18,19].

One of the most important features of cancer cells is their ability to undergo an unlimited number of cell divisions, which provides them with immortality. This is due to the autocrine secretion of growth factors, which results from the cancer cells’ independence from external factors. In addition, cancer cells can interact with the surrounding normal cells, creating a specific microenvironment as well as stimulating nearby normal cells to produce growth factors [20]. Cancer resistance to apoptosis results from various defense mechanisms, but the most common cause is mutations leading to the loss of p53 protein function [18]. It is also characteristic for cancer cells to increase the expression of telomerase, whose role is, among others, to restore telomeres, which allows cells to divide in an unlimited way and avoid replicative senescence [18].

## 5. Telomerase—Key Factor in Immortality

Telomerase is expressed in more than 85% of cancers [21,22]. Human telomerase is a ribonucleic complex consisting of several subunits, but two of them are considered to be critical and are required for full activity in vitro: hTERT and hTERC (human telomerase reverse transcriptase and the human telomerase RNA component). The molecular weight of the holoenzyme exceeds 500 kDa [3]. The catalytic protein subunit hTERT is encoded by the hTERT gene located in locus 5p15.33. The hTERC unit is an RNA template for DNA synthesis and is encoded by a gene located in locus 3q26 [23]. It forms a structure called the “glove structure”, which allows telomerase to wrap around the end of the chromosome and synthesize new telomeric repeats. While the hTERC gene is expressed in most tissue types, the hTERT gene is limited to certain groups of tissues, including stem cells, activated lymphocytes, embryos, cancer, and cancer stem cells [21]. In vivo, the entire complex contains additional proteins such as NOP10, NHP2, GAR, and dyskerins that bind to hTERC and stabilize the entire complex [24].

The mechanism of telomerase action can be divided into three basic steps: primer recognition and binding, the elongation of the DNA strand, and enzyme translocation or dissociation. In the first step, the sequence of the telomeric DNA strand located at the 3’ end of the chromosome and the sequence of the matrix RNA of the enzyme are matched complementarily. In the next step, the hTERT subunit elongates the DNA strand, using the RNA fragment as a template. Then, the telomerase moves towards the 3’ end of the newly synthesized chromosome to add further repeats or dissociates from the telomere [3].

## 6. Canonical and Noncanonical Functions of Telomerase

The main function of telomerase is to provide immortality to cells by maintaining the integrity of the chromosome ends. This RNA-dependent DNA polymerase synthesizes telomeres by reverse transcription and eliminates the end replication problem as well as prevents chromosomes from fusing. Consequently, telomerase has been recognized to have a huge role in cancer metabolism, aging, and degenerative diseases. The function of telomerase associated with the altering telomere metabolism is called the canonical function of the enzyme [3,21].

However, the role of telomerase (or its subunits) is not limited to telomeric functions. It also contributes to the regulation of key metabolic mechanisms in cells, such as gene expression, mitochondrial metabolism, signal transduction pathways, and stress responses (Figure 1). These nontelomere functions are called noncanonical telomerase functions [21].

The hTERT subunit is associated with key signaling pathways that control proliferation, migration, and cell differentiation during embryonic development or carcinogenesis, among others [21]. One of these is the Wnt/β-catenin pathway, in which hTERT acts as a cofactor in the β-catenin transcription complex by binding to BRG1. Thus, hTERT is involved in the regulation of gene expression, including cyclin D1, MYC, and AXIN2 [25,26]. Recent studies indicate that the role of hTERT is also to stabilize MYC protein and regulate the binding of MYC to target promoters. Thus, telomerase contributes to the activation or repression of MYC family genes [26,27].

Telomerase is also involved in the NFκB signaling pathway by regulating inflammatory signals in cancer cells. NFκB can be regulated by various factors, including phosphatases, kinases, cytokines, and lncRNAs, that modulate the expression of proinflammatory genes [28,29,30,31]. Telomerase (hTERT) can mediate these processes by interacting with transcription factors or chromatin-modifying factors [21]. Together with the p65 subunit of NFκB, hTERT can bind to gene promoters in the presence of an inflammatory stimulus. Studies suggests that the p65 subunit is the main regulator of proliferation, while hTERT enhances its function [32]. In addition, the inhibition of telomerase activity reduces the binding of p65 to the promoter sequence. Therefore, it is suggested that the expression of p65-dependent genes in the NFκB signaling pathway is linked to the expression of hTERT [28,32].

It is known that telomerase can protect cells from oxidative stress, which is associated with a reduced level of reactive oxygen species (ROS). However, due to conflicting findings, these relationships are still not fully understood [33,34,35]. The result of oxidative stress is the presence of 10–20% of the cellular hTERT pool in mitochondria. The oxidative-stress-induced overexpression of hTERT in mitochondria has been shown to limit UV-induced mtDNA damage and mutagenic agents (e.g., ethidium bromide) and to protect mitochondrial DNA by reducing the cellular production of superoxides and ROS [21,35,36]. Other studies reported impaired mitochondrial function due to the reduced expression of hTERT, which manifested in increased or decreased membrane potentials, changes in the mtDNA copy number, and altered mitochondrial mass and ATP levels [37,38,39]. An association between the mitochondrial localization of hTERT and the drug resistance of cancer cells was also observed [38,40,41]. All these results, although still not fully understood, point to a complex relationship between mitochondria, telomerase, and ROS [21,35].

It was shown that hTERT is not the only subunit that exhibits extratelomeric functions. The hTERC subunit is also able to control the expression of other genes, as was demonstrated during studies on a mouse melanoma cell line. In this study, the suppression of mTERC (mouse TERC) resulted in the inhibition of the expression of more than 100 genes. Some of them were involved in the glycolytic pathway, indicating a possible relationship between hTERC and cancer metabolism [42]. Studies in human cell lines suggest that the upregulation of hTERC could lead to the inhibition of angiogenesis and the downregulation of metastasis-related gene expression [43]. It has also been shown that hTERC is involved in the transcription of genes mediated by the telomerase-dependent NFκB signaling pathway [32,44].

Telomerase subunits may have many different nontelomeric functions. Many discrepancies can be seen in the results of studies on their noncanonical functions. This is probably a result of the diversity of research models and the lack of appropriate experimental tools to gain a thorough understanding of the mechanisms involved in cells. Moreover, in cancer cells hTERT is a weakly expressed protein, and the existence of alternative splice variants of hTERT complicates the interpretation of the results [35].

## 7. Therapeutic Potential of Telomerase Subunits

### 7.1. Telomerase as a Target for Cancer Therapy

In most somatic cells, telomerase activity is below the detection limit [3,45]. The exceptions are embryonic stem cells, marrow stem cells, intestinal crypts, hair capsules, testicular cells, and ovarian cells [46,47,48]. Most fetal tissues, apart from brain cells, show high telomerase activity at 16–20 weeks of development [49]. However, it was not detected in cells from a 2-month-old baby, with the exception of reproductive cells [49,50]. High telomerase activity is characteristic of most cancers, especially advanced and metastatic cancers [45]. Normal breast tissue has been shown to lack telomerase activity [48]. In the case of noninvasive ductal carcinoma, telomerase activity was demonstrated in 75% of examined samples, and for ductal or lobular carcinoma this value was 88% [48]. Due to the high expression of telomerase in tumor cells, it has become an important biomarker of cancer and a promising therapeutic target [45].

The idea of telomerase-targeted therapy is to selectively induce the apoptosis of cancer cells with minimal side effects of the therapy on normal cells. To achieve this goal, the following therapeutic strategies are being pursued: the direct inhibition of telomerase, the inhibition of hTERT or hTERC promoters, strategies based on telomeres, and the development of vaccines based on the telomerase protein complex elements/antigens. Several strategies directed against telomerase, e.g., imetelstat, Vx-001, and GRNVAC1, are currently at various stages of clinical trials, which suggests that in the coming years a drug targeting telomerase might be included in the pharmacopoeia [4,51,52,53,54].

One of the biggest problems in modern oncology is the lack of drug sensitivity of cancer cells, mainly due to acquired resistance to anticancer agents. Studies indicate a link between telomerase inhibition/repression and the increased sensitivity of cancer cells to certain drugs, e.g., doxorubicin, 5-fluorouracil, and cisplatin [55,56,57]. The mechanisms of telomerase resistance vary depending on the type of cancer and the type of drug used, which means that they are still not fully understood. Nevertheless, telomerase-targeted therapy shows great promise and is being tested in a wide range of cancer models [22].

One of these approaches is based on the direct inhibition of telomerase’s enzymatic activity. There are currently several compounds that cause direct telomerase inhibition. The first and probably the best-studied is GRN163L, also known as imetelstat. It is a 13-mer oligonucleotide that inhibits telomerase by antagonistically binding to the telomerase RNA template. Preclinical studies have demonstrated high efficacy of GRN163L [4,45,58]. For example, the treatment of breast cancer cells with this drug resulted in the inhibition of telomerase activity and progressive telomere shortening, leading to a reduced risk of tumor formation and cell invasiveness [58,59]. Imetelstat showed an inhibitory effect on telomerase activity in CSC (cancer stem cell) populations of HER2+ breast cancer cells. Moreover, imetelstat, both alone and in combination with trastuzumab, reduced the CSC fraction and inhibited CSC functional ability, as shown by decreased mammosphere counts and invasive potential. These results strongly suggest that the addition of imetelstat to trastuzumab may enhance the effects of HER2 inhibition therapy, especially in the CSC population [60]. Unfortunately, in clinical trials, imetelstat did not achieve the expected efficacy and showed strong toxic effects, mainly neutropenia and thrombocytopenia [4]. Despite this, there are still high hopes for this drug, and more clinical trials are underway.

Telomerase inhibitors have shown limited improvement in patient prognosis. This is related to the fact that, despite telomerase inhibition, cancer cells are still viable, and some of them acquire resistance to inhibitors through ALT (alternative lengthening of telomeres) induction. It is suggested that a much more effective strategy would be to use the hTERT or hTERC promoters to express cytotoxic factors in a cancer-tissue-specific manner or to generate mutations in hTERT promoters to reduce its expression [4,45].

Simultaneously, gene editing technology based on CRISPR interference and programmable base editing seems to be a very useful approach and might validate activating TERT promoter mutations as a cancer-specific therapy. Li et al. exploited CRISPR to downregulate TERT expression in glioblastoma cells. Introduced modifications blocked the binding of E26 transcription factor family members to the TERT promoter, reduced TERT transcription as well as TERT protein expression, and induced cancer cell senescence and proliferative arrest [61]. 

### 7.2. Targeting Telomeres

Much attention has been given to strategies in which telomeres are destabilized while telomerase activity is not affected. This approach has the advantage of therapy efficacy, even in cancers that maintain telomeres through an alternative telomere (ALT) elongation mechanism. Three main categories of potential drugs affecting telomeres can be distinguished: G-quadruplex stabilizers, tankyrase inhibitors, and T-oligo [4].

One of the functions of the G-quadruplexes is to protect the ends of the chromosomes from the action of exonucleases [62]. To enable telomere elongation, they are partially unwound by telomerase [63]. The G-quadruplex stabilizers prevent the access of the enzyme to the substrate and thus prevent telomere restoration [45,64]. Importantly, this may be efficient in ALT-positive cancer cells as well by conferring antiproliferative properties to the cells [64,65]. However, due to the diversity of G-quadruplex structures, there are difficulties in determining the conformations of the structures under physiological conditions, which makes drug design much more complicated [65]. Currently, several formulations are under investigation, including TMPyP4, BRACO-19, and telomestatin, but they have not yet entered the clinical trial phase, unlike CX-5461, which is currently in phase I clinical trials, with promising results [4,66,67,68].

Another category of drugs affecting telomeres is tankyrase inhibitors, which prevent telomerase from binding to telomeres, thereby inhibiting their elongation. In addition, they also show nontelomeric functions such as affecting the β-catenin-pathway-dependent transcription pathway [4,69,70]. Tankyrase inhibitors were shown to cause more rapid telomere shortening, which could lead to cell death [70,71]. These include, e.g., IWR1, XAV939, and JPI-547, which are currently in clinical trials [4,72]. 

Another compound that is still under investigation is T-oligo. It is an 11-mer oligonucleotide that is homologous to the 3′ end of the telomere, which, when introduced into cancer cells, increases p53 activity and induces autophagy and apoptosis, contributing to the elimination of cancer cells [4,70,73]. This therapeutic strategy induces DNA damage responses (DDR) in several cancers, e.g., melanoma, breast carcinoma, and lung cancer [74,75,76,77]. Interestingly, T-oligo has little or no effect on most normal cells [70,76,78]. Initial studies suggested that T-oligo did not affect telomerase activity [78,79]. However, recent studies indicate that the action of T-oligo may include a reduction in telomerase expression [73,75].

### 7.3. Genetic Profiling in Different Cancers

In 2013, Huang et al. [80] and Horn et al. [81] reported high frequencies of hTERT promoter mutations in melanoma. Following their discoveries, some further reports showed varying distributions of these mutations across cancers and indicated hTERT promoter mutations to be the most frequent noncoding mutations in cancers. [82,83]. As demonstrated, somatic mutations in the regulatory regions of target genes (transcription modulation) also affected telomere length [84,85,86,87]. Consequently, hTERT mutations were indicated to be important factors in oncogenic activation, cancer stemness, and proliferation [86]. Importantly, breast cancer has a very low frequency of TERT promoter mutation, which implies epigenetic regulation in this type of cancer and possibly a wider systemic effect, especially in HER2-positive breast cancer [88]. 

### 7.4. Immunotherapy

A novel approach is to use telomerase as a target for the development of immunotherapy, and two main strategies have been adopted. The first is based on direct in vivo immune activation with a vaccine directed against hTERT, which sensitizes cells of the immune system to tumor cells [4,51]. Telomerase is degraded by proteasomes, leading to the formation and presentation of hTERT-derived peptides as antigens on the cell surface [4,51,89]. This induces an antitumor response of CD8+ or CD4+ cytotoxic T lymphocytes [51]. Promising vaccines include GV1001 and Vx-001 (Geron, Foster City, CA, USA), both of which effectively stimulate the immune system while inducing minimal side effects, as demonstrated in clinical trials [4,53,90,91]. Another strategy is the ex vivo activation and expansion of immune cells using the GRNVAC1 vaccine. The patient’s isolated dendritic cells are transfected ex vivo and then administered to the patient via intradermal injection. This induces a strong immune response with only grade 1 toxicity [4,51,91]. Importantly, these studies have already progressed to in vivo experiments, which makes this strategy less elusive and more realistic [92,93,94].

Another peptide-based cancer vaccine consisted of two human leukocyte antigens. The subcutaneous injection of the TERT (572Y) peptide followed by the subcutaneous administration of the TERT (572) peptide was aimed to elicit a specific and possibly optimal cytotoxic T-cell (CTL) response against hTERT-expressing tumor cells [95]. The new trend in cancer immunology is based on nucleic acids and an adjuvant approach. As shown in 2007, chemokine adjuvant strategies might enhance the tumor-antigen-specific (hTERT-targeted) immunity induced by vaccines. The hTERT DNA vaccine consisted of a plasmid containing the carboxyl terminal end of the TERT (cTERT) gene encapsulated in multilayered liposomes with a hemagglutinating virus of Japan coating. The study demonstrated that CCL21 administration before cTERT DNA vaccination significantly augmented tumor-antigen-specific immunity against breast cancer [96]. 

Other studies on hTERT-based immunotherapy (three phase I/IIa clinical trials covering malignant melanoma, non-small-cell lung cancer, and prostate cancer) showed that UV1, a second-generation telomerase-targeting therapeutic cancer vaccine, triggered highly dynamic and persistent telomerase-peptide-specific immune responses that lasted up to 7.5 years after the initial vaccination. The superior immune response kinetics were observed in the melanoma study [97].

### 7.5. New Generation Vaccines

Recent studies showed an optimized DNA plasmid encoding an inactivated form of hTERT, named INVAC-1, capable of triggering cellular immunity against tumors. The intradermal injection of INVAC-1 followed by electrogene transfer (EGT) in a variety of mouse models elicited broad hTERT-specific cellular immune responses, including in highly CD4+ Th1 effector and memory CD8+ T cells. Furthermore, therapeutic INVAC 1 immunization in an HLA-A2 spontaneous and aggressive mouse sarcoma model slowed tumor growth and increased the survival rate of 50% of tumor-bearing mice. INVAC-1 vaccination was safe, well-tolerated, and immunogenic when administered intradermally at the three tested doses in patients with relapsed or refractory cancers. Disease stabilization was observed for the majority of patients (58%) during the treatment period and beyond [98]. 

Altogether, there are nine ongoing hTERT-immunotherapy-based clinical trials at different stages, out of which two are being carried out in solid tumors, including breast cancers [99]. These are: NCT02960594—hTERT Immunotherapy Alone or in Combination With IL-12 DNA and NCT02301754—INVAC-1 Anti-Cancer hTERT DNA Immunotherapy. As hTERT is reported to affect the tumor metastatic potential, there is a need for an appropriate model to study this phenomenon. One of the models that reflects the in vivo conditions in humans is a mouse breast tumor model based on 4T1 tumor cells. The 4T1 mammary carcinoma is a transplantable tumor cell line that is highly tumorigenic, invasive, and, unlike most tumor models, can spontaneously metastasize from the primary tumor in the mammary gland to multiple distant sites, including the lymph nodes, blood, liver, lung, brain, and bone. Moreover, the progressive spread of 4T1 metastases to the draining lymph nodes and other organs is very similar to that of human mammary cancer. Another advantage of 4T1 is its resistance to 6-thioguanine, which enables the precise quantitation of metastatic cells [100].

Despite being considered a promising antitumor strategy, to date, cancer vaccination has not been shown to be a clinically useful approach in patients with severe disease [101]. This overall negative result is at least in part due to the poor effectiveness of gene transfer with the currently employed strategies and, particularly, the frequent onset of a neutralizing antibody response against the viral vectors, which limits the use of repeated vaccinations [102,103]. Therefore, new immunization regimens that are able to reduce the neutralizing antibody response and maintain an active immune response over time are needed to assess the actual potential of cancer vaccination in clinical practice. The results of a recent study suggested the safety and feasibility of V934/V935 hTERT vaccination (an adenoviral type 6 vector vaccine expressing a modified version of hTERT, administered alone or in combination with V934, a DNA plasmid that also expresses the same antigen) in cancer patients with solid tumors. Further investigation is needed for a proper evaluation of the efficacy of this strategy and its potential impact on clinical practice [103].

## 8. Summary

Intensive research on the regulation of telomerase in cancer cells in recent years has allowed for the understanding of the mechanisms accompanying tumor proliferation, telomere elongation, and noncanonical enzyme functions. All of these make it possible to develop new, effective anticancer drugs. Unfortunately, there are still many unsolved questions concerning telomeres and telomerase, such as whether telomerase expression has oncogenic properties [51]. It is also important to note that it has still not been possible to develop an effective telomerase-based drug that would be approved for clinical use. This shows the need for further research. However, it is undeniable that the study of telomerase has contributed to a deeper understanding of cancer biology and has opened up promising new therapeutic avenues [4,51,104]. Research conducted in recent years has led to the development of many structurally diverse compounds with different mechanisms of action. Therapies based on targeting telomerase and telomeres or combination therapies may prove to have high efficacy against telomerase-positive tumors while sparing neighboring normal cells [45,105]. In addition, the use of molecularly targeted therapy makes it possible to bypass the side effects characteristic of conventional therapies [45,73]. Hopefully, the knowledge accumulated so far will soon allow researchers to improve the therapeutic efficacy and eliminate the side effects of previously known drugs targeting telomeres and telomerase or will allow the development of new and effective drugs. However, further research is needed to make this possible [105]. 

## Figures and Tables

**Figure 1 ijms-23-12844-f001:**
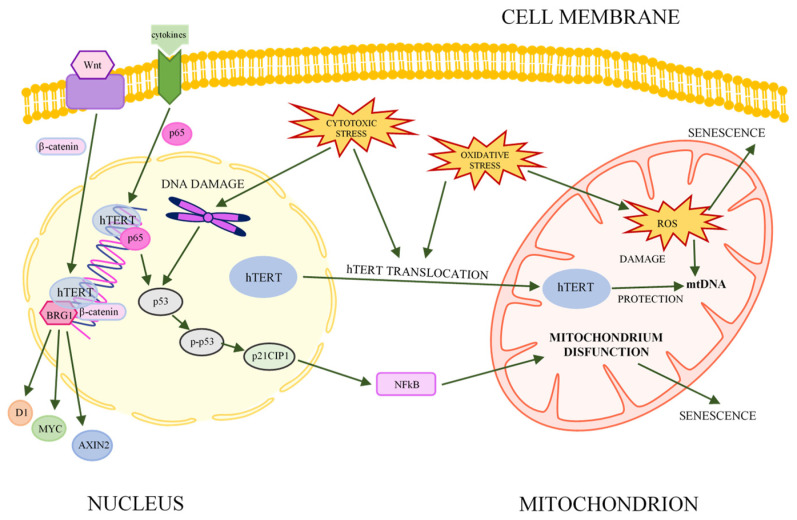
Association between hTERT, key signaling pathways, and senescence (acc. [21,22,39], modified). The hTERT subunit of telomerase can be involved in various metabolic pathways (e.g., NFκB, Wnt/β-catenin, and adhesion as well as senescence) and affect the expression of various genes. Under the influence of stress, hTERT is translocated to mitochondria, where hTERT protects mitochondrial metabolism by lowering the level of ROS and binding to mtDNA.

## Data Availability

Not applicable.

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
