# Peer review of "The Role of Telomerase in Breast Cancer’s Response to Therapy"

_ijms, 2022, doi:10.3390/ijms232112844_

Round 1

Reviewer 1 Report

In this manuscript Judasz et al. review the role of telomerase in breast cancer. I think such a review could be informative, however, this manuscript will need considerable expansion and restructuring to deliver that goal.

Specifically, the authors did not integrate “breast cancer” and “telomerase/telomere” well. Though the review starts with breast cancer, most of the writing is focused on telomeres/telomerase, and no clear efforts were made to connect the two. For instance, was Imetelstat tested in breast cancer, and what was the outcome? What is the rationale for targeting telomerase or telomeres in breast cancer in particular? 

The authors also fail to review some of the most exciting advances in telomere/telomerase. In 2013, Huang et al. and Horn et al. reported high frequencies of TERT promoter mutations in melanoma (both in Science). Following their discoveries, Killela et al. surveyed many cancer types for these mutations (PNAS, 2013), and showed varying distribution of these mutations across cancers. Importantly, breast cancer has very low frequency of TERT promoter mutation. Later, genomic analysis showed TERT promoter mutations are the most frequent non-coding mutations in cancer (Fredriksson et al. Nat Genet, 2014).

Many works have been done along the line. For instance, Barthel used sequencing data to look at telomere lengths (Barthel, et al. Nat Genet, 2017); Sieverling et al. studied genomic fingerprints of telomere repeats (Sieverling et al. Nat Commun 2020); Noureen et al. computationally predicted telomerase using a new signature (Noureen et al. Nat Commun 2021). A complementary pathway, ALT, was mapped to ATRX/DAXX mutations (Heapy et al. science). New mechanistic insights were also reported, including Chiba et al. Science, 2017; Stern et al. Cell Rep, 2017; Genes Dev, 2015; Bell et al. Science, 2015.

The binding of GABP factors to TERT promoter mutations can be disrupted by CRISPR, creating a therapeutical opportunity to target the cells with this mutation. See Li et al. Nat Cell Bio, for example.

The authors should make an effort to incorporate these new advances and shift the focus of the review to telomeres/telomerase rather than breast cancer. Of course, if they feel compelled to review telomerase in breast cancer, that is also fine, provided a better integration of the two concepts.

Author Response

First, we would like to thank the reviewer for a thorough and constructive review.

As recommended, we slightly modified the title into its present form, i.e., “The Role of Telomerase in Breast Cancer Response to Therapy.”

The whole manuscript was reconstructed and supplemented with chapters devoted to immunotherapy and genetic profiling to complete the broad view of the role of telomerase as a therapeutic target in breast cancer. The justification for the model selection was also provided.

We also made an effort to incorporate new advances in the field and shift the focus of the review to telomeres/telomerase rather than breast cancer with better integration of the two concepts.

All amendments are marked up using the “Track

Changes” function. 

Reviewer 2 Report

The manuscript "The role of telomerase in breast cancer resistance to therapy" presented by Eliza Judasz et al. contains a review of the actual literature on the properties of TERT and the progress of TERT-related anti-tumor therapies. Still, several points need to be addressed before accepting it for publication:

The main point of concern is that the current title of the manuscript is misleading. The reviewed literature is mainly devoted to the general properties of TERT and TERT-related therapies and lacks specific points regarding the role of TERT in breast cancer development and treatment. Namely, if the current title remains as is, the manuscript would require an additional review of a number of papers from multiple groups investigating TERT expression and TERT vaccine candidates testing in murine 4T1 adenocarcinoma cells.

There is also a small number of mistakes in the text, including plural for "characteristics" in line 44, lack of ALT abbreviation decoding after its first use in line 223 etc. Minor editing is advised.

The manuscript is recommended for publication upon minor revision.

Author Response

First, we would like to thank the reviewer for a thorough and constructive review.

As recommended, we slightly modified the title into its present form, i.e., “The Role of Telomerase in Breast Cancer Response to Therapy.”

The whole manuscript was reconstructed and supplemented with chapters devoted to immunotherapy and genetic profiling to complete the broad view of the role of telomerase as a therapeutic target in breast cancer. The justification for the model selection was also provided.

We also made an effort to incorporate new advances in the field of anti-hTERT-vaccine research and shift the focus of the review to telomeres/telomerase rather than breast cancer with better integration of the two concepts.

All amendments are marked up using the “Track Changes” function. 

Round 2

Reviewer 1 Report

The authors have improved the manuscript.